# Reuse and Retrofitting Strategies for a Net Zero Carbon Building in Milan: An Analytic Evaluation

Daniela Besana *[iD] and Davide Tirelli [iD]

Department of Civil Engineering and Architecture (DICAr), University of Pavia, 27100 Pavia, Italy
* Correspondence: daniela.besana@unipv.it; Tel.: +39-0382-985404

**Abstract:** The building sector accounts for 38% of carbon emissions, the principal cause of climate change. To meet the targets set by the Paris Agreement, including zero net emissions by 2050, it is necessary that governments develop a culture of sustainability. Whole Life Carbon Assessment of a building, comprehensive of operational and embodied carbon (EC), is described by EN15978:2011. Net Zero Carbon Buildings (NZCB) achieve a balance of zero emissions during their life cycle, promoting both reduction and compensation by adopting many strategies (e.g., reuse of existing structures, design for adaptability and disassembly, circular economy principles). Choosing bio-based materials is also helpful to compensate for EC, thanks to the biogenic carbon stored during their growth. The aim of this research is to find out which strategies are relevant to meet NZCB target, in order to apply them to a case study of reuse of an abandoned building in Milan, highlighting the design process. Material quantities were extracted from the BIM model and imported in OneClick LCA to assess embodied carbon emissions (A1–A5 modules, material production and supply, transport, construction). The final design stage achieved a reduction of 91% in EC compared to a standard new construction, while the bio-based materials compensated for the rest. Further research can improve the Environmental Product Declaration (EPD) of materials and assess the entire building life cycle.

**Keywords:** Net Zero Carbon Building; retrofit; embodied carbon; sustainable design; LCA; carbon emissions; circular economy; reuse; EPD; bio-based materials

## 1. Introduction

This contribution establishes the methodological premises starting from the climatic urgency that places the world in front of the need to understand how to intervene. In the construction sector, the main causes of global warming are greenhouse gases with the consequent increase in the global average temperature which results, for example, in the growth of hot days number, in the rise of sea levels and unusual meteorological phenomena of strong intensity.

In order to control the increase in temperatures, it has been shown that the most effective solution is to eliminate greenhouse gas emissions ($CO_2$e) by 2050. It would mean going from about 51 billion tons of $CO_2$e, emitted on average every year, to zero grams, a very complex challenge that can only be pursued by working simultaneously on all production sectors [1]. One of the most relevant documents in the scientific and political debate on climate change is the report provided by the Intergovernmental Panel on Climate Changes (IPCC), Sixth Assessment Report, August 2021. The hypothesized future scenarios vary on the overall $CO_2$e emissions depending on the socio-economic path taken, the best of which considers reaching the zero emissions target towards 2050 and subsequently implementing a carbon negative strategy, as shown in Figure 1.

In all cases the rise would still exceed the target temperature of 1.5°C, reaching 1.6 °C during the 21st century and possibly hitting a peak of 4.4 °C in the worst simulation, which would lead to a sharp increase in extreme events, including floods, fires, monsoons and heat waves [2].

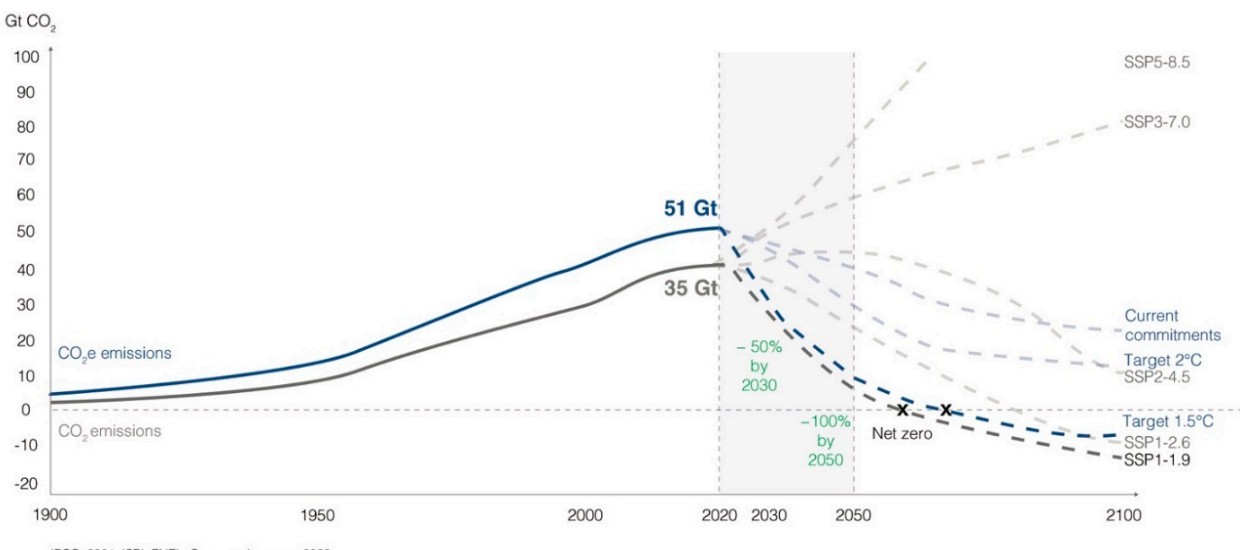

**Figure 1.** Greenhouse gases emissions: after the rise until 2020, there are different possible pathways for the future. Each path implies direct consequences for the increase of temperatures [2–4].

The theme of environmental protection and the reduction of anthropogenic impact has been growing since the nineteen-seventies, when the first effects of climate change began to be recorded. The fundamental stages are here summarized with the purpose of framing aims and methods. At the United Nations Conference held in Stockholm in 1972, the issue of safeguarding against risks to the environment was addressed, exploiting technological innovation, scientific research and education [5].

Subsequently, in 1987, the concept of sustainable global development was defined with the Brundtland report, then resumed with the Rio conference in 1992 (Agenda 21), the United Nations Millennium Summit in 2000 (Millennium Development Goals, MDGs), the World Summit on Sustainable Development (WSSD, Rio + 10) in 2002 and the United Nations Conference on Sustainable Development (UNCSD, Rio + 20) in 2012 [6], noting every time the urgency to pay attention to the problem of pressure of man on the natural system, of its vulnerability and resilience. However, many of the initiatives undertaken can be classified as weak sustainability, without a concrete benefit for the environment. The international cooperation programs culminated in 2015 with the drafting of the 2030 Agenda [7], divided into 17 principles of the Sustainable Development Goals (SDGs) and 169 targets, which take up the previous Millennium Development Goals of 2000 to update them and make them more adherent to the challenges of the following decade.

The Paris Agreement, signed in 2015 during COP21 by 197 countries, is currently the most recent and broadest agreement, and it sets climate protection as its primary objective, committing itself to adopt the solutions necessary to maintain the global average temperature rise below 2 °C and closer to 1.5 °C by 2050 [8]. Comparing the progression in the programming of international civil rights and environmental protection summits with the temperature trend emerges that the objectives set have become more and more restrictive, but emissions have grown constantly.

To meet the targets set by the Paris Agreement, zero net emissions in 2050, it is therefore necessary that governments work together through an intense development of clean technologies and the production of energy from wind and photovoltaic plants, hydroelectric system and nuclear fission [9]. At the same time, it is necessary to raise awareness and develop a culture of sustainability in the population in order to change choices and habits such as, for example, increasing the installation of heat pumps and virtuous behavior.

The construction sector is currently one of the main emitters of carbon dioxide, whose contribution is estimated at around 38% of total emissions in 2019. Of this percentage,

28% is attributable to buildings, while 10% is due to industries that produce materials and building components: a total of about 10 billion tons of $CO_2e$, a figure higher than that of transport and the rest of industrial production [10]. Buildings are also one of the main consumers of electricity, accounting for 33% of the entire production in 2020, a percentage destined to rise by 2050 to 72% at the European level. In Italy, over 57% of electricity production still takes place from the combustion of fossil fuels, primarily natural gas, contributing significantly to emissions [3,4,9] (Figure 2).

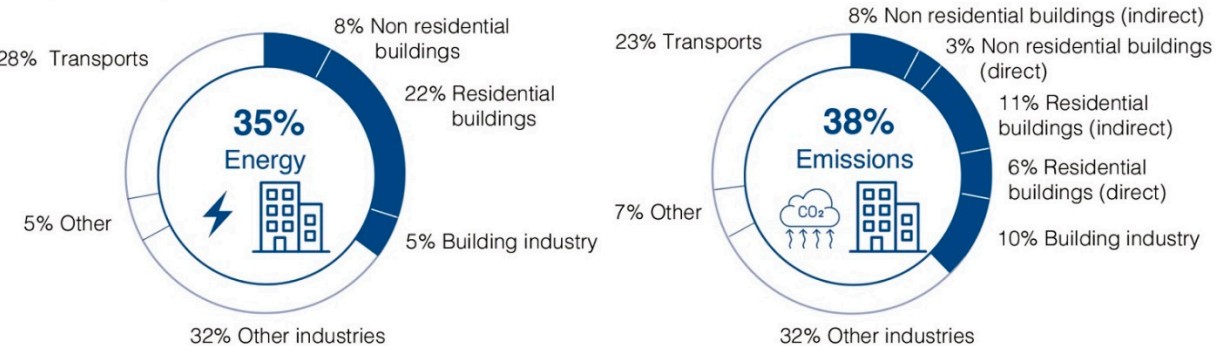

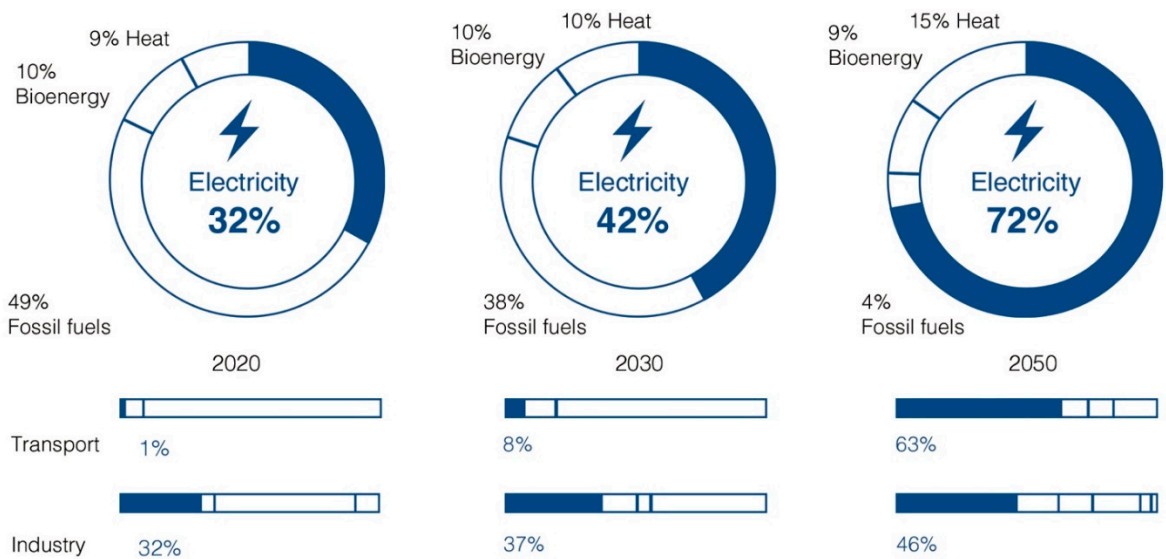

**Figure 2.** (**a**) Percentage of energy consumption and emissions of the building sector, 2019 [10]; (**b**) increasing demand of electricity by the building sector: in 2050 it will be crucial to produce enough electricity to supply all the sectors, as it will be the main source of power [3].

The design of existing and future buildings is, therefore, a key point in addressing the environmental problem. According to the forecasts of the United Nations Framework Convention on Climate Change (UNFCCC) it would be necessary to reach the net-zero target in operation for all new or refurbished buildings by 2030 and ensure a reduction of at least 40% of the embodied carbon, while by 2050 all buildings will have to meet the net-zero carbon requirement over their entire life cycle. With this perspective, it would be possible to live in healthy, accessible, inclusive and convenient places, ones suitable for the target of an additional 1.5 °C [11].

The construction sector will be able to face transformation over the next three decades by applying three concepts: efficiency, sufficiency and decarbonization [12]. The development of cleaner concrete and steel, as far as possible, will bring benefits but the greatest efficiency should come from the extension of the useful life of the materials, associated with a more careful design that limits their use.

To certify the actual performance of the building in terms of environmental sustainability, some evaluation methods have been established in order to have a reliable and shared tool for certifying the quality of the building heritage. The two most famous standards are the Building Research Establishment Environmental Assessment Method (BREEAM) and Leadership in Energy and Environmental Design (LEED©), followed by others applied in more limited geographical areas. Although every building intervention leads to an anthropogenic action on the natural environment, it is still possible to minimize the impact with the use of certain construction processes that comply with the requirement of net zero carbon, or an overall balance with zero greenhouse gases emissions during the life cycle of the building.

The authors' contribution is aimed at developing the refurbishment project of an abandoned building in Milan by quantitatively evaluating the impact of the embodied carbon associated with the intervention, to reach the Net Zero Carbon Building (NZCB) goal, following a specific design process that integrates net zero carbon strategies. Starting from the meta design project phase it is necessary to establish the criteria for the choice of materials but also the planning of the intervention on the existing building as well as drafting a digital model to highlight where the emissions come from.

This should allow the designer to define a quantitative and applicative evaluation to satisfy the zero emissions goal, ready to be applied in the execution of the refurbishment.

## 2. Materials and Methods

The Net Zero Carbon Building (NZCB) condition is obtained "when the amount of carbon emissions associated with a building's embodied and operational impacts over the life of the building, including its disposal, are zero or negative" (UKGBC definition) [13].

To achieve the NZCB goal it is possible to adopt many strategies, both in the case of new construction and in the case of retrofitting or refurbishment.

Firstly, the design of an NZCB requires a different cultural and methodological training on aspects that should not be neglected. This paper wishes to summarize the manner in which an NZCB is defined, the methodology to achieve this target and how to assess it according to what has already been published in literature, then applying some of the described strategies to a case study in Italy, where the Net Zero design approach has not been experimented with.

The paper firstly analyzes the methodological and theoretical context (e.g., reuse and retrofit strategies, Whole Life Carbon Assessment methodologies, NZCB strategies) and secondly focuses on the specific case study in Milan, evaluating the reuse capability and the WLCA and NZCB strategies applied.

### 2.1. Reuse and Retrofit Strategies

To deal with a process of reuse of the built heritage, the preventive evaluation of residual resilience is of great importance to define thresholds that do not compromise "the material consistency" and "the safeguarding of cultural identity values" [14]. The purposes of a redevelopment intervention can be limited to a retrofit of the interior spaces and the envelope, to improve accessibility and MEP equipment or make the building more efficient from an energetic point of view.

Aiming at the reduction of embodied carbon and soil consumption, the "reuse and retrofit" and "build less, build clever" approach strategies [15] are the most useful to reduce greenhouse gas emissions, allowing up to 50% emissions saved compared to a new construction intervention, mainly due to the reduction in the demand for materials such as steel and concrete.

On a scale of embodied emissions, it can be observed how the "build less" solution is actually valid: there would be no related embodied emissions if the building is used with no modifications, as opposed to cases of light retrofit (e.g., energy requalification of facades), hard retrofit or reuse with the addition of new volumes and, lastly, demolition with reconstruction. (Figure 3)

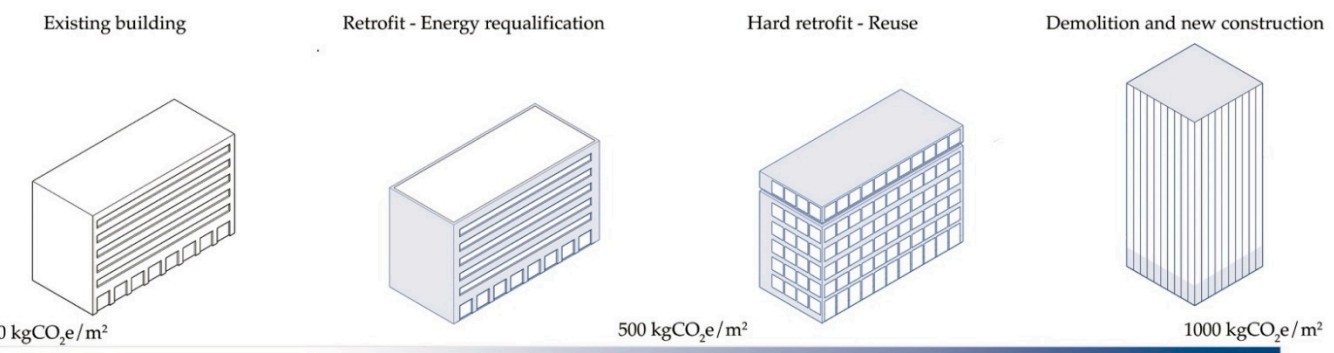

**Figure 3.** Increase in embodied carbon emissions proportionally to the importance of the intervention.

The advantages of this more drastic choice, however, are high in terms of compositive and organizational free layout. Before planning the reuse of a building, in fact, it is advisable to proceed with an evaluation of the performance and the potential for reuse (i.e., PAV, Performance Adequacy and Vulnerability and RTE, Resilience Threshold Evaluation), because some functions may be incompatible with the existing building and needing partial or complete demolition; for example, sanitary buildings require compliance with very specific need for adaptability and internal distribution of flows, which may not be satisfied by existing structures [14–16].

Most of the buildings built between the 1950s and 1970s suffer from poor architectural and construction quality, leading to three phenomena of obsolescence to which a reuse and refurbishment project must answer: technological obsolescence, due to the insufficient performance of the components inadequate to contemporary needs; functional obsolescence, due to modification of the location needs of certain functions, such as industrial buildings in the urban fabric; obsolescence of the image, due to the poor design quality in origin given that in the real estate world the design of the facades has a very important weight.

The main reason, however, remains energy efficiency aimed at reducing consumption in the construction sector, which is responsible for about 35% of energy consumption and 38% of greenhouse gas emissions [10]. The interest in these retrofit interventions is also supported by the administration offering specific bonuses and financial support (such as Superbonus 110% for energy efficiency approved in Italy with the decree DL 34/2020); this is also one of the main goals of the Italian National Plan for Recovery and Resilience (PNRR), set with the M2C3 mission [17]. The retrofit thus becomes an opportunity to improve building efficiency and the quality of life for the occupants, with a lower cost and a lower environmental impact: a process that follows the principles of sustainability [18].

From a compositional point of view, there are different approaches: volumetric addition, volumetric subtraction and wrapping (Figure 4).

The volumetric addition strategy can be carried out starting from the superposition, a very widespread intervention method thanks to the freedom offered in positioning the new volumes inside the shape, overhanging, adjacent and leaning against the existing, altering the initial appearance in a significant way, also from a structural point of view. The main advantages of adding on the roof are the transformation of the host building without increasing the footprint on the ground and ensuring the reversibility of the intervention [19].

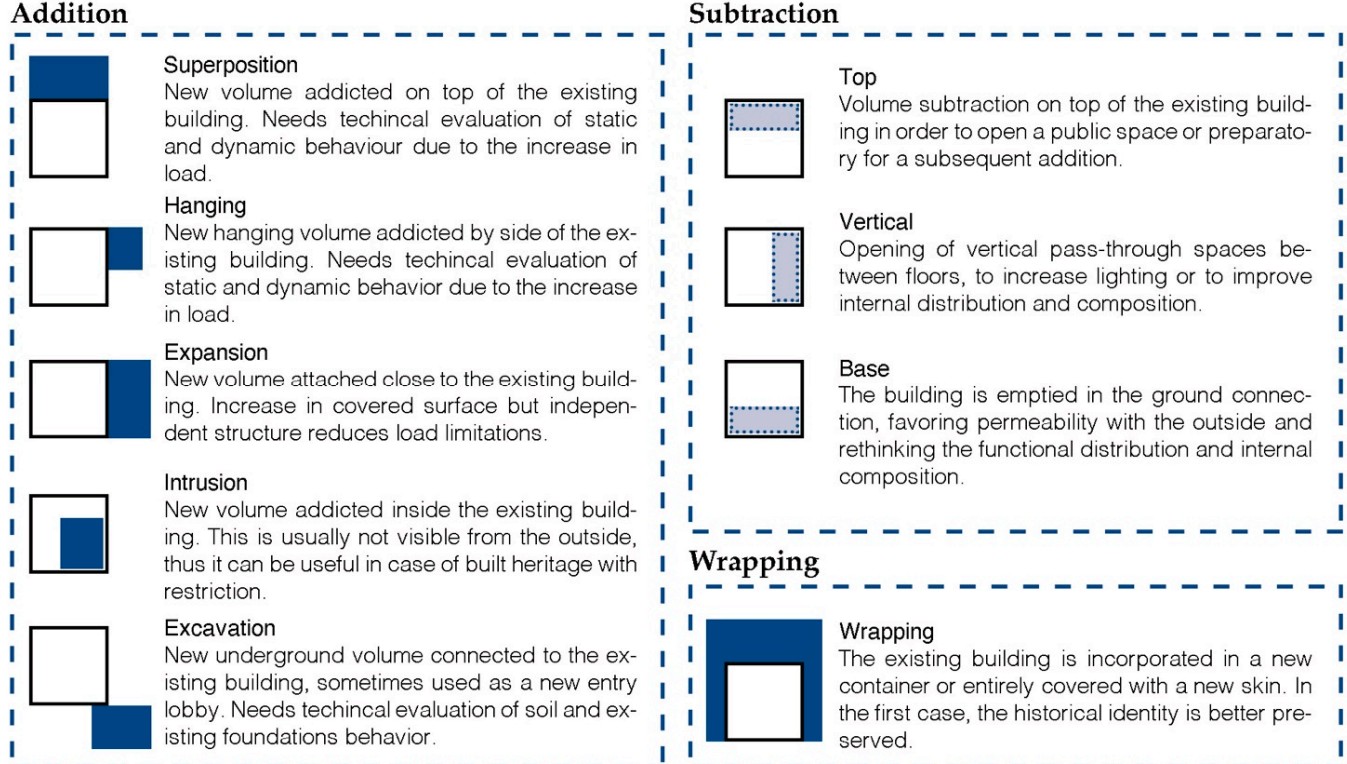

**Figure 4.** Volumetric strategies appliable in case of building reuse; it is possible to sum up in three different groups.

The addition strategy can also be implemented with suspended additions, which can be structurally dependent or independent or adjacently added, or new volumes leaning against and connected to the building.

The intrusion (box in a box), on the other hand, exploits the pre-existence, usually characterized by large openings or large empty volumes, to insert new parasitic bodies that thus increase the available space and colonize it, possibly without appearing from the outside. Another solution that can be adopted to increase the volume is the excavation outside or inside buildings to add volumes to the base, after checking the feasibility and existing foundational structures. Also in this case, the added volumes could be totally invisible from the outside or be declared with elements that catalyze attention, as in the case of the Louvre Museum in Paris (projects by Pei, Bellini) [20,21].

The subtraction strategy removes part of the host building to allow the insertion of new functions to meet the new requirement framework, for example the opening of double-height spaces, opening of loggias or portions of the base or the top floor. Often the strategy of subtraction is preparatory to the addition of new volumes (e.g., intrusion or superposition), thus obtaining an absolute result identical or greater than zero [19].

The wrapping strategy consists in covering the building with a new skin. The result can be obtained by applying a new adherent skin, working on the thick envelope with benefits also from a bioclimatic point of view (recladding), or by incorporating the existing building in a new container, enhancing its artistic value [22].

Retrofit interventions, aimed at meeting new requirements or performances not foreseen in the original project, can affect the body or the skin of the building and are mainly aimed at energy efficiency (Figure 5).

The advantage of working on existing buildings is the possibility of testing the initial thermal, hygrometric and functional conditions and of working on a real-scale model to implement the various design solutions [23]. Body strategies allow interventions on the existing structure and on the non-structural components (vertical and horizontal closures) with the aim of making the building passive and exploiting the benefits offered by sun

exposure and natural ventilation. For example, the adoption of passive solar systems, with direct gain or solarium, allow solar radiation to enter the rooms through the glass surfaces to be captured by high thermal inertia elements (accumulation masses) such as the structural parts or the floors [24]. This strategy can be integrated with volumetric addition or subtraction through glazed volumes that benefit from the greenhouse effect [25].

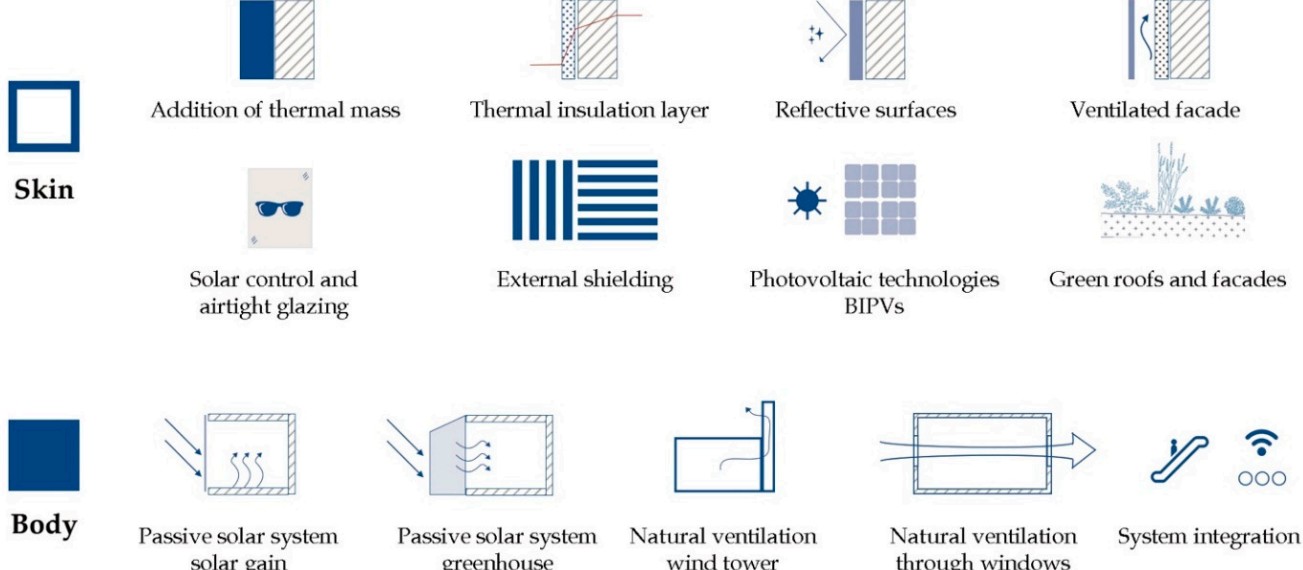

**Figure 5.** Retrofit strategies appliable to existing buildings: skin solutions and body solutions.

The interventions on the envelope tend mainly to reduce solar gains or intercept them to produce energy and heat, as well as activate micro-ventilation processes. It is possible to increase the thermal masses of the opaque surfaces of the external envelope, for example with a massive new cladding, or by providing insulation with thermal insulators often integrated into ventilated facade systems. To reduce solar gain and control in glass windows especially during summer season it is possible to add in the cladding external shields, fixed or movable. The presence of a planted layer on the roof (intensive or extensive green roof depending on the thickness and water demand) or on the façade increases the thermal mass and reduces the thermal flow between inside and outside. [25]

Building Integrated Photovoltaics (BIPV) technologies are another possible solution to renew external cladding while producing clean energy [26].

*2.2. Carbon Emissions Assessment*

The assessment of greenhouse gas emissions is a crucial point in defining the decarbonization strategy to be applied in each production sector. To achieve this purpose in construction, it will be necessary to pay attention, in addition to what is already required by current regulations, to the design process and to the performance evaluation of the emissions deriving not only from the use of the building (operational carbon), but deriving from construction, maintenance and the end of life of the building (embodied carbon) [27]. Focusing on the operational aspect leads to neglecting the amount of embodied carbon due to make a building passive so that, in some cases, it can even exceed the operational contribution during the life cycle [28].

The built-in carbon contribution from construction can exceed 50% of overall emissions, of which 70% comes from just six items of materials or components (concrete, steel, aluminum profiles, glass and services such as lighting, heating, cooling) and the maintenance phase can contribute 20% of emissions over the life cycle [29]. It therefore appears important to assess emissions throughout the life cycle of the building according to the Whole Life-cycle Carbon Assessment (WLCA) methodology, also considering the end of

life through three possible scenarios, from the most inefficient to the optimal: demolition, recycling, reuse [30]. Positive signs are the greenhouse gas emissions produced during the life cycle, which are currently unavoidable to build a building, even if they can be reduced substantially; with a negative sign, the contributions that lead to the absorption and offset of emissions are recorded (see Section 2.4). The result obtained can be a value greater than or equal to zero or even less than zero, achieving the goal of having a carbon neutral or carbon negative building.

The most widespread and shared method for the WLCA assessment is currently the one proposed by the EN 15978: 2011 standard, which divides emissions into five phases: production (phases A1–A3), construction (A4–A5), use (B1–B7), end of life (C1–C4), beyond life (D), to be assessed separately (production, construction, use, end of life, beyond life) (Figure 6) [29]. The portion that can be classified as embodied carbon is the largest, while the operational carbon is represented by modules B6 and B7 [31].

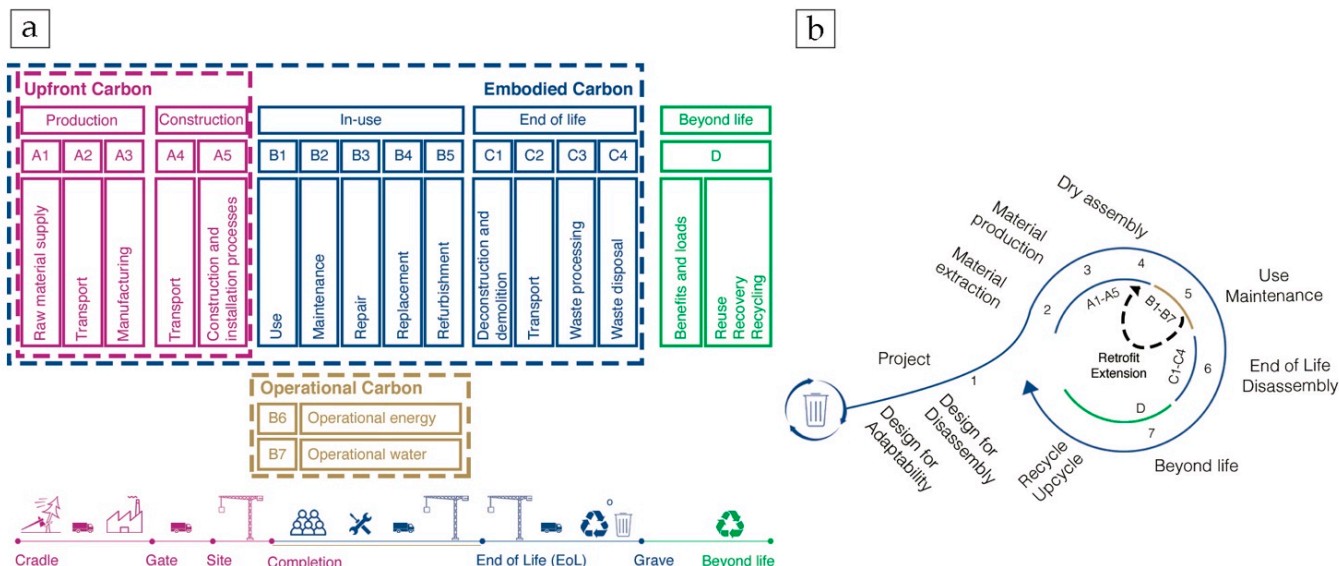

**Figure 6.** (**a**) Carbon emissions classification based on EN 15978:2011: each module refers to a specific phase in the life cycle of the building; (**b**) circular economy principles applied to the building process, aimed at extending its life.

Standard buildings score high carbon emission, both embodied and operational (Business as Usual case). A study carried out by the World Business Council for Sustainable Development (WBCSD) and ARUP, after analyzing the building emissions benchmarks published by some research institutes and the calculations carried out on buildings designed by ARUP, highlighted how the embodied carbon of a building in the phases A1–A5 (upfront carbon, cradle-to-gate) is approximated in 1.000 $kgCO_2e/m^2$ (BaU) and for the entire life cycle (phases A–C) in 1.300 $kgCO_2e/m^2$ (BaU). This amount is expected to decrease by 40% by 2030 to comply with the decarbonization process [29].

The elements that make up the building (envelope, structure, foundations, internal partitions, plant engineering) have a different useful life, therefore in the calculations of embodied emissions, generally referring to fifty years, it is necessary to evaluate the replacement of some components.

In general, for the foundations a life cycle of one hundred years can be estimated, for the vertical structures fifty years, for the facades and the rest of the envelope thirty years, for technological systems and internal partitions ten years. Given that many buildings are demolished before fifty years, some elements still have potential for later reuse [32].

The operational carbon of a building is instead approximated to 70 $kgCO_2e/m^2/year$ (BaU), equivalent to 220 $kWh/m^2/year$ of energy consumption. Assuming a useful life of the building equal to fifty years, the operational emissions (B6–B7) correspond to

3500 kgCO$_2$e/m$^2$ [29]. In Italy it is mandatory starting from 1 January 2021 according to the fulfillment of the nZEB requirements for new or redeveloped buildings, whose average energy consumption is significantly lower than that reported in the Business as Usual scenario. According to the reference climatic zone, consumption varies, but on average for a new residential building in Lombardy about 55 kWh/m$^2$/year are used for heating and cooling [33], which would correspond to about 17.5 kgCO$_2$e/m$^2$/year, a reduction of 75% from the previous condition. Over the life cycle of the building, therefore, there would be greenhouse gas emissions of about 1000 kgCO$_2$ e/m$^2$.

The legislation also provides that the data are as up to date as possible and geographically close to the site taken into consideration.

From this point of view, the lack of Italian databases can constitute a limitation on the truthfulness of the counts carried out in the WLCA assessment.

The WLCA evaluation has been made only for a small minority of projects in the world so far, mainly due to the lack of databases and of standard procedures and shared by professionals as well as insufficient carbon literacy [34]. Many states are proceeding to form public databases with nationally valid ranges of values, such as in France (INIES), the Netherlands (milieudatabase.nl), Belgium (EPD Database), Germany (oekobaudat.de), the United States and the United Kingdom (ICE, Inventory of Carbon and Energy).

The ideal application at the executive level involves the use of the data provided in the Environmental Product Declaration (EPD) of each material included in the project, since the manufacturer has carried out precise calculations of the greenhouse gas emissions on its life cycle on the model of the European directive EN 15804; in this way there would be a maximum level of detail for the analyzed building.

For the analysis of the case study, a software for Life Cycle Assessment, One Click LCA (Helsinki, Finland), was used to perform a preliminary calculation of the embodied carbon produced in stages A1–A5 during the definition of the design choices: using the actual quantitative data provided by the EPD certificate of each product, it was possible to minimize the value by substituting the heavier materials in the following stages of calculus [35].

### 2.3. Net Zero Carbon Buildings Strategies

The decarbonization process of a building can be summarized in four phases: reduction of embodied carbon, reduction of operational carbon, increase of renewable energy supply, compensation of residual embodied emissions [13–15] (Figure 7a).

The reduction of emissions and of energy and materials consumption is essential to achieve the goal: saving on emissions helps to be able to zero emissions through offsetting. It is important to ensure that the development of net zero carbon practices guarantees a competitive advantage in economic terms; for this reason the UKGBC proposes three guidelines within its framework:

- Pay-per-pollute: the cost of offsetting emissions is proportional to the emissions produced; therefore, the adoption of good emission reduction practices is encouraged for economic reasons.
- Encourage transparency: making public the data on detected and not estimated greenhouse gas emissions generates a virtuous cycle that drives efficiency, also raising public awareness on the issue.
- Encourage action in the present to be able to tighten the links in the future: starting with a net zero carbon approach for the operational and construction part helps in the task of evaluating the entire life cycle in the future, when the buildings are decommissioned, when there will be more knowledge about it [13].

To reduce embodied carbon, a likely solution is to reuse existing buildings and to carefully evaluate the contractor's requirements, following the slogan "Build Less, Build Clever" [15]. The design for adaptability and design for flexibility principles are also of great importance to extend the life cycle of the building, helping to insert new function in

the future. For example, new volumes should have generous story height and very regular structures located on the perimeter to allow for free internal distribution [36].

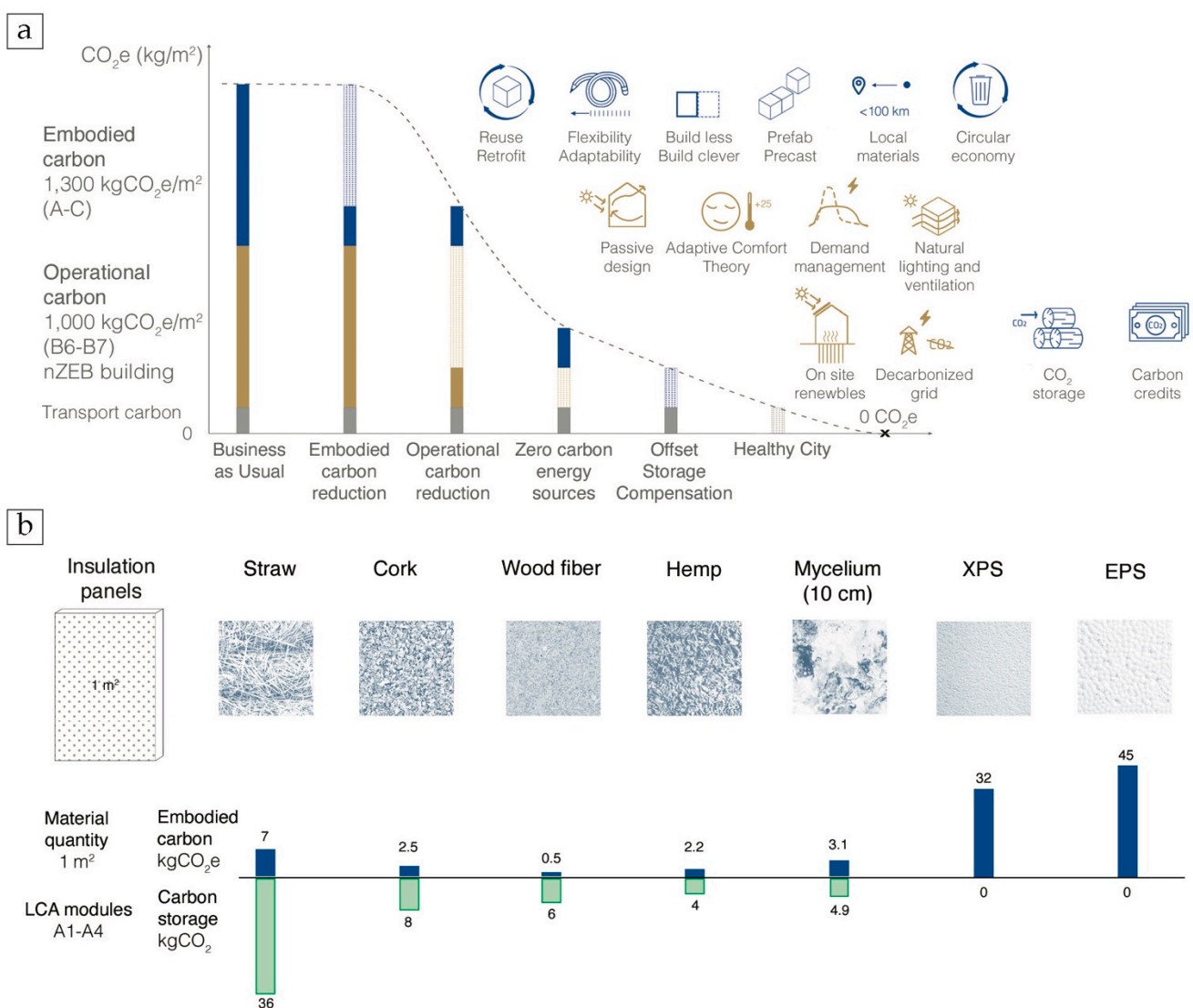

**Figure 7.** (**a**) Strategies for building decarbonization. It is possible to get the net zero target in four steps starting from the Business as Usual condition: reduce embodied carbon, reduce operational carbon, use decarbonized sources for the operational energy, and store and compensate for the remaining embodied carbon. If the building is located in a healthy city, travelling to and from it can be decarbonized easily. (**b**) Comparison between mineral based and bio-based insulation materials, evaluating the carbon footprint and the carbon stored in each material. Data sourced from [36,37]; Campioli, A., et al. "Design Strategies and LCA of Alternative Solutions for Resilient, Circular, and Zero-Carbon Urban Regeneration: A Case Study". In *Regeneration of the Built Environment from a Circular Economy Perspective*; Della Torre, S., Cattaneo, S., Lenzi, C., Zanelli, A., Eds.; Springer: Cham, Switzerland, 2020. https://doi.org/10.1007/978-3-030-33256-3_20; CINARK, "The construction material pyramid", https://www.materialepyramiden.dk/# (Available Online); OneClick LCA EPD database (available in software). Color palette key description: blue for embodied carbon, gold for operational carbon, green for stored carbon.

Preferring local materials is helpful to reduce transportation carbon emissions, as well as the use of circular economy principles, both for material supply during construction

and maintenance stage as well as at the end of life of the building, favoring recycled and recyclable materials [38]. Circular economy principles are also related to design for disassembly, that is possible mainly with the prefabrication and dry technologies [32].

To compensate for the remaining embodied carbon, it is possible to adopt bio-based materials which offer a considerable advantage in the level of stored carbon; in Section 2.4 these materials will be discussed more widely. Purchasing carbon credits can be a faster solution to compensate embodied carbon, but it is closely linked to weak sustainability.

To reduce operational carbon emissions, it is possible to adopt solutions for passive design, such as natural ventilation and lighting, passive heating and cooling and adaptive comfort theories: the opening of large suitably screened windows allows passive heating indoor spaces in winter and cooling in summer by natural ventilation [24].

The demand for energy for heating and cooling of the building is closely linked to the needs of the occupants, who tend to prefer to be able to intervene directly. Otherwise, users will be less satisfied, not being able to intervene if conditions are uncomfortable; this leads people to adopt behaviors that increase consumption (e.g., the "blinds down, lights on syndrome") [23–36,38,39]. It has also been documented since the 1980s that users are much more willing to tolerate situations of less than ideal thermal and lighting comfort if they understand that they depend on natural conditions (adaptive comfort theory): for example, people using natural light adapt to conditions that does not respect the standard luminance values, whereas they won't tolerate artificial lighting that doesn't meet their expectations [40]. The use of a demand management system allows for the management of when energy is used, shifting demand away from peak periods in which supply emissions are at their highest level, for example, with load reduction or installing batteries. This also contributes to the reduction in the size of the plants to make their use more efficient when they operate at low speeds (i.e., most of the time) [15].

To balance the remaining needs of energy without emitting operational carbon it is possible to produce electricity and heat with renewable sources, such as photovoltaic plants or geothermal pumps, while the remaining energy need that has to be purchased from the grid will be progressively produced in a cleaner way, thanks to the decarbonization of electricity planned for the next decades [3].

Regarding emissions due to the movement of occupants, which do not strictly depend on the building, the reduction of car travel for employees and visitors must be encouraged promoting both the use of public transport and soft mobility, following healthy city principles; furthermore, the transition between internal combustion engine to electric vehicles is already reducing the carbon footprint for mobility [41].

### 2.4. Negative Carbon Contributions

The possibility of reducing greenhouse gas emissions in architecture is closely related to design choices and materials used, both for the structure and for the finishes and insulation. The problem can be addressed by reducing the emissions produced, avoiding their release into the atmosphere, and by absorbing carbon dioxide, and therefore having a negative weight within the WLCA calculations; however this solution is not shared by the entire scientific community, as mentioned later.

The use of mineral-based materials that derive from industrial processes has been widely employed in construction in the last century, while the adoption of cement and steel has gradually imposed itself, assuming that they were the best technologies, supported by great economic advantages, replicability, reliability and resistance. However, these technologies not only have little chance of being decarbonized in the production cycle because they are highly energy-intensive, but they are also difficult to recycle once used, with great waste of raw materials [36].

Therefore, the transition to an architecture built with materials of biological origin (bio-based materials), engineered in the twenty-first century to obtain the required values of reliability, resistance and cost-effectiveness, is the challenge to be accepted in order

to initiate a reconciliation between the pressing request for new built volumes and the environmental issue.

In Europe alone, the construction of buildings for about two billion square meters is planned in the decade 2020–2030, which will make up 60% of the building stock viable in 2060 [42], while by 2050 the world population should increase up to ten billion people [43]. Consequently, the emissions necessary to build new infrastructures could amount to 60% of the total carbon budget compatible with the limit of temperature increase of 2 °C; the more we will be able to start the ecological transition in the building stock in this decade, the more will be the environmental benefits [44].

The most promising organic materials for applications in construction are wood, hemp, cork, straw, mycelium and, to a lesser extent, cellulose and wool, all of which adhere to circular economy processes and are able to subtract and store $CO_2$ during their life cycle like "carbon sponges" [37,45] (Figure 7b).

However, the scientific literature has not expressed an unambiguous consensus on the method of calculating these negative contributions, as they could lead to misunderstandings if it was not specified in which module of the building's life cycle they occur.

Organisms can store carbon during their lifetime, but sooner or later it will be re-emitted during the decomposition process. For this reason, the first approach, called the "0/0 approach", neglects the positive and negative contributions along the life cycle, assuming that sooner or later they will compensate; the second, called the "−1/+1 approach", keeps track of all the biogenic carbon exchanges, in and out, for each module of the life cycle, guaranteeing a higher and more precise level of information. However, this method can be misleading if applied only for module A, as negative contributions can, for example, lead to the result of a carbon negative building, and then disprove this hypothesis by also analyzing the subsequent modules B, C and D where the positive biogenic contributions occur [46].

Researchers have shown that carbon stored in buildings can offset the temporary reduction of carbon stored by forests until trees are grown again: if sourced from controlled producers, wood could be a sustainable solution [47]. However, the time of regrowth of the forests, and therefore the restoration of the ability to absorb carbon dioxide, may be too long to make the wood a carbon neutral product, unlike straw or hemp which have significantly higher regrowth rates, quickly restoring the ability to sequester $CO_2$ from the environment [48]. For these reasons, the adoption of biogenic materials in construction is certainly a good practice, but it is better not to rely too much on stored carbon and the negative contribution in the WLCA assessment because of the uncertainty and volatility of these contributions.

*2.5. LCA Software Implementation*

LCA and WLCA analysis can be performed with the help of software that can relate materials, EPDs and quantities deriving from the design and construction to calculate the impact during the life cycle of products or buildings; in particular, there are national and international tools such as Elodie (France), Totem (Belgium), Tally, USai, GaBi or OneClick LCA [49]

Many tools can work independently as separate tools, while others can be integrated within the Building Information Modeling (BIM) environment, sourcing data directly from the model; in fact, the BIM environment creates a holistic approach to the building from cradle to grave, enabling the division of information smoothly for a multi-disciplinary approach.

The BIM model is important in both cases to speed the LCA assessment, since exporting a bill of quantities (BOQ) which contains a list of materials with their respective quantities becomes easy and reduces the risk of errors. The materials' quantities have to be linked to the chosen LCA profiles, for instance, Global Warming Potential (GWP), to obtain the embodied carbon emissions of each item.

The LCA tool thus can run the analysis and evaluate the environmental impact, in order to visualize the results. [49]

The strategies of BOQ export and then to import into the LCA tool, or loading the model directly imported into the LCA tool, are both convenient. In the second case it is likely to be more time-efficient since every material will be paired directly to its EPD, both in case of specific and generic materials. As mentioned before, using generic materials reduces the precision of the assessment, especially for the type of energy used in the manufactory of the specific country or the distance of the manufacturing plant to the building site [50].

Another possible implementation of LCA and BIM design is with the digital twin (DT) of the as built stage, a tool that can help to find out the best solution during the maintenance phase according to the LCA analysis and to control key decision variables that define the NZCB [51].

For the case study, a BIM model using Autodesk Revit was made to find out material quantities in the BOQ, which was exported into a spreadsheet. Then the data were imported into the One Click LCA software manually, associating each material with those included into the software catalogue, each one with its EPD. The Level of Development (LOD) of the Revit model for the case study was LOD 300, given the geometric definition of the building itself and specific objects displacements, quantities and dimensions, allowing a bigger quantity of information relevant for the BOQ. The One Click LCA software was chosen thanks to its simplicity yet completeness that were previously appreciated in other LCA evaluations.

A table shows the building components involved in the project and an estimate of the quantities obtained from the project technical drawings, as well as the distance and mode of transport of the various components. These values have been increased by 10% to have a safety margin aimed at including potential exceeding quantities in the executive phase. The calculation also includes the energy and water consumption data linked to the construction phase, as well as the quantities of materials deriving from the demolition that will be transported out of the construction site to be recycled, incinerated or disposed of in landfills.

For each material or component, the unitary embodied carbon quantity (GWP indicator), expressed for areal or volumetric quantity, is multiplied by the corresponding dimension; thus, each result is summed in order to find out the overall embodied carbon of the building. The same calculus applies to the biogenic carbon quantities, which are summed without the 10% increase in material dimensions to be conservative with the global results.

### 2.6. Existing Building Description and Evaluation of Reuse Capability

The building chosen for the reuse project hosted the headquarters of the Provveditorato agli Studi di Milano, located in Ripamonti Street, on the border of the Porta Romana rail yard, one of the areas in Milan that will be most actively transformed during the next decade [52]. The building has an area of approximately 10.000 m$^2$ (GFA) distributed in two main bodies: an office linear block and a slab; it was used by the institution from 1987 to 2007 and abandoned since then. The structure is made of reinforced concrete, with low stress rates (approximately 40%) and very regular structures, with modest spans in the office block and large spans in the plate: a favorable condition for their reuse and the addition of new volumes on the roof.

Given its location in a vibrant regeneration area and its flexibility, the building appeared to be suitable for a reuse project, focused on the redevelopment of the entire complex to host a research center, multi-tenant offices and public spaces.

Data about the building have been provided by the owner in June 2021, ENPAM Real Estate, assuring sufficient information about the construction phase that occurred in the 1980s.

The building offers good chances of reuse given its regular structure and big internal heights, two key points for reusing buildings [15].

In fact, the office building is built with a regular concrete framework, organized in three building units with a 3.20 m span and a 3 m net internal height, while the plate, originally thought viable for industrial production, but was then used for storage, a conference center and a canteen, is made of precast concrete structure with big span (13 m × 9 m) and a substantial internal height (about 8 m). These dimensional characteristics, together with the large sections of the vertical structural elements, made it possible to hypothesize the reuse of the office building without changes since the requirement framework has not changed, with the addition of a wooden rooftop and new public functions for the plate, similar to other case studies such as Toni Areal in Zurich, Switzerland, or Fondazione Prada [53] and Hangar Bicocca in Milan, Italy.

For the plate, where the structural sections are larger but must bear a greater distributed load given the large spans, an analysis of the loads acting in the vertical direction was carried out and showed that the stress level on the pillars is now approximately 40%. This allows the addition of parasitic volumes on the roof [54], without structural changes, and in the analyzed case, to add a new five-story tower made of wood, minimizing structural reinforcements and new foundations, as will be discussed in Section 3.2 (Figure 8). A structural pre-dimensioning has been done to test the feasibility of the addition strategy.

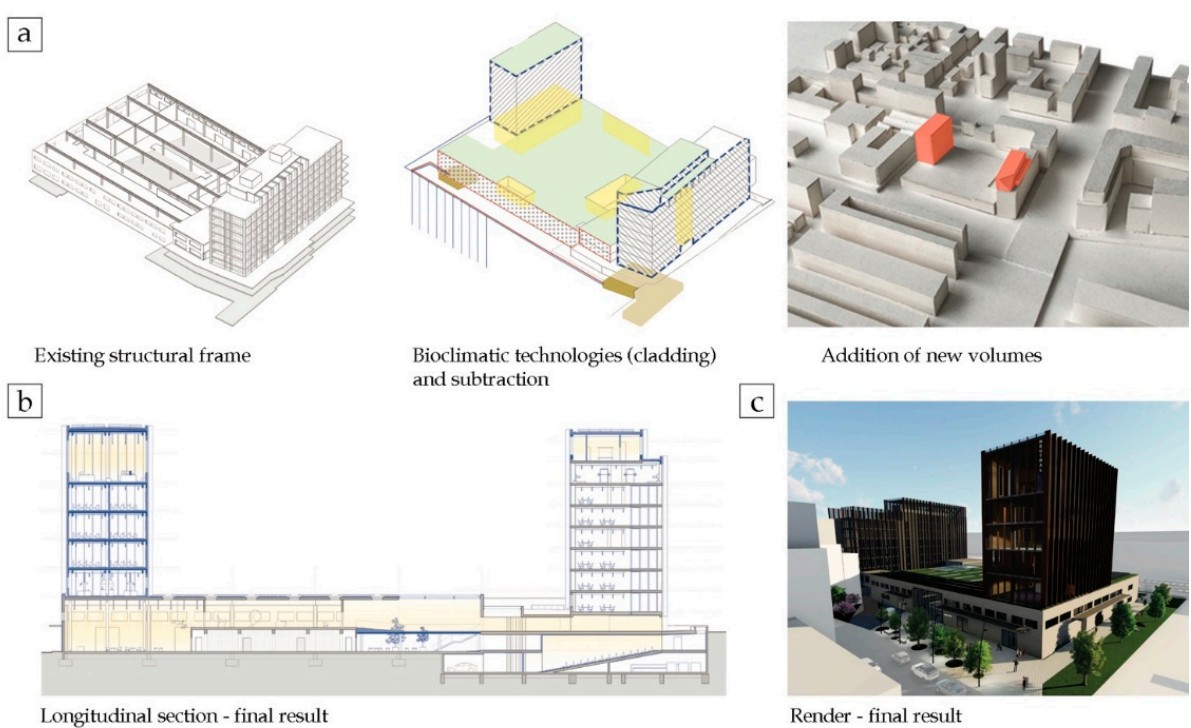

**Figure 8.** (**a**) The current status of the building shows a very regular structural frame; starting from this status, demolitions and volume removal are represented in yellow while volume additions on top of the building are highlighted in red; (**b**) section of the building at project status, blue lines highlight additions and new elements; (**c**) render view of the project status, the building is converted into a research center; note the new superposed volumes made in wood.

On the other hand, the plate's structural frame cannot bear horizontal loads (e.g., of seismic origin) because of the plane disposition without appropriate bidirectional trusses; the project must consider this limitation.

The existing building is still divided into different areas, making it possible for future users to accommodate different functions or to let the offices to different tenants without heavily changing the layout; still, the division between plate and offices is rigidly present; thus the project tries to open the interstitial spaces (following the "in-between" theory of Hermann Hertzberger) [55] (Figure 8).

## 3. Results

### 3.1. Ripamonti 42 Building Reuse: Design Process

The project plans to reuse the existing property according to a conservative approach, aimed at a redevelopment addressed to avoid the three obsolescence phenomena described in Section 2.1. The choice to include a research center about the development of new materials and offices for startups requires different types of spaces, also with a certain amount of flexibility and adaptability over time. The existing building can meet these requirements given the great variety of coexisting building types; the office block can resume its role with minimal changes; the plate, initially conceived as an industrial building, can accommodate functions that need double heights or larger spaces, such as laboratories; the conference center, housed in a big, stepped room, can be transformed into an exhibition space open to all.

The project aims to be a Net Zero Carbon Building, following the strategies exposed in Section 2.3, while reuse and retrofit main approaches were applied early in the design process (Section 2.1). All three reuse strategies are applied in the case study to meet the new requirements: addition of a new parasitic volume on the plate (superposition), increasing new surface for didactics and laboratories of the research center and on the office building to host new rentable surfaces; subtraction in the plate for natural ventilation and lighting of internal spaces; and vertical subtraction in the office building to open new loggias.

At the same time, the retrofit strategies involve both overall aspects, such as the thermal mass given by the existing concrete structures, and detailed aspects related to the building's envelope. The combination of the two actions leads to significant impacts on the entire architectural complex, both from a functional and aesthetic point of view; for instance, the design of the cladding of the facades of the existing block and of the new tower was particularly careful to integrate the double glass skin, functional to activate natural micro ventilation processes, with vertical fins (and horizontal on the south front) in acetylated wood to reduce the radiation entering the building, giving overall lightness and transparency to the volumes. The positioning of the fins was developed to divide in half each span of the original concrete structure, left exposed after the (minimal) demolition operations, mainly related to the disassembly of the existing spandrel facade and the masonry of the attic floor, which is aligned with the facade.

These new facades are fundamental in the management of natural ventilation (summer cooling) and for the passive gain solar system with activation of the thermal masses (winter heating). Removing unnecessary finishing layers and increasing the size of the windows thus allows sunlight to penetrate and heat the concrete masses by radiation. The logic is extended to the large openings made in the slab and, to a lesser extent, in the new tower, which despite being structurally made of wood has natural lime mortar screeds that increase the thermal mass.

During the meta-project and project phases the reduction of embodied carbon associated with the intervention was carefully considered, defining the choice of materials and technologies. A preliminary calculation of the embodied carbon was performed using the software One Click LCA during the entire design process and then made definitive at the end by using quantitative data of materials and elements exported from the Revit BIM model, allowing it to reduce its value as much as possible by substituting the heaviest materials in terms of greenhouse gases emissions. The bill of quantities (BOQ) was then imported into One Click LCA software, pairing each material with those in the software catalogue. The aim to use local materials and components, produced as nearly as possible to the construction site, is not always reflected into the calculus, since many materials are offered in the catalogue as generally valid for Europe or other European countries, while Italian products are still scarcely included in the database. Then, the software showed an assessment for the embodied carbon emissions due to A1–A5 modules of the production life cycle and the consumption of electricity, water and energy, complete with graphs for clearer data visualization.

*3.2. Analytical Discussion on Embodied Carbon Results Obtained with the Software*

The software releases a report certifying the performance in terms of embodied carbon of the building, comparing the absolute value (t $CO_2$e) with the project square footage (Gross Floor Area (GFA), sum of Gross Area and Accessory Area) and the intended use. For the intervention in via Ripamonti 42, a value of 89 kg $CO_2$e/m$^2$ was obtained, which ranks in class A for reuse interventions for cultural purposes (A1–A5) [56], and a value of 82.5 kg $CO_2$/m$^2$ for biogenic carbon.

The value of 89 kg $CO_2$e/m$^2$ is very low when compared with the averages for new constructions and retrofits, i.e., 500 kg $CO_2$e/m$^2$ in the case of reuse and redevelopment and 1000 kg $CO_2$e/m$^2$ for new constructions (upfront carbon) [29]; the project achieves a reduction of 82% and 91% respectively. The main credit of such a drastic reduction in emissions is due to the almost complete reuse of the structures and the large surface area considered (17.750 m$^2$ GFA), on which, however, few modifications and additions were sufficient. In particular, the use of materials such as concrete and steel is limited to a few interventions, including the foundations of the new tower which have been specially designed as a plate of the minimum possible size, only 100 sqm. Three design solutions for addition were considered in order to choose the best fitting in terms of flexibility, structural feasibility and carbon impact (Figure 9).

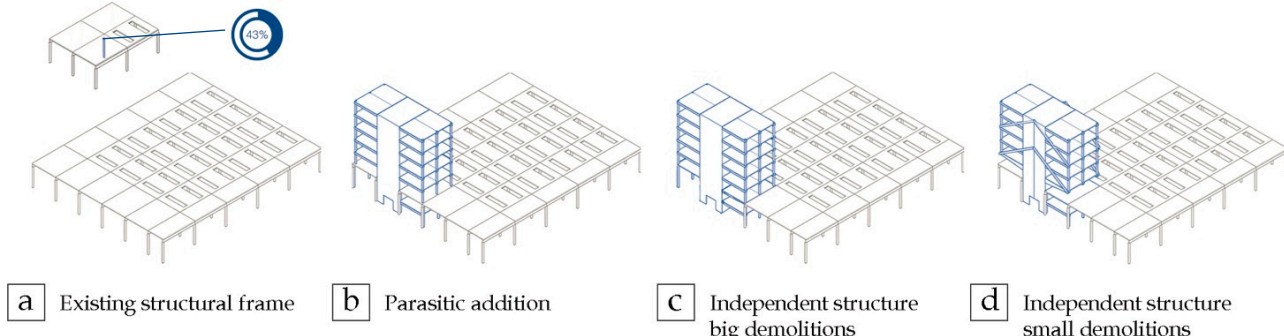

**Figure 9.** (**a**) The existing plate structure is loaded to 43% of its limit resistance, so new load can be added on top of it; (**b**) parasitic addition with a tower made of structural wood: this solution minimizes the necessity for new foundations, limited only to the attachment of new wooden walls for vertical connections in one span; (**c**) demolitions of three spans of the existing structures and constructions of new wooden tower: this solution brings more demolition waste and three times bigger new foundations; (**d**) demolition of one span and new wooden and steel structure, not bearing on the existing concrete frame: this solution has low demolition waste but greater complexity and less adaptability.

Given the choice of using biogenic materials for the structure, the facades and the insulation, the building can store a large amount of carbon, further mitigating the environmental impact and getting closer to the target of carbon neutrality: carbon stored by the materials is equivalent to 82.5 kg $CO_2$ e/m$^2$. This figure, calculated according to the "−1/+1 approach", however, does not consider any positive emissions due to the end of life of the materials (module C) [46]. This is the same reasoning applied for module D (Beyond Life) linked to the recycling and reuse of components, which is usually neglected as the hypothesized benefits may not materialize and are delayed over time [48]. It was therefore considered more correct to specify the value without making a sum for clarity in the presentation of the results, aware that the abatement of the embodied carbon emissions must be envisaged above all with other strategies that avoid emissions at the origin rather than compensate them.

The analysis of the embodied carbon is organized by the One Click LCA software dividing the emissions by material and by technological systems of the building system, as shown in Appendix A. In particular, the first part is about the construction materials and the second one is about construction site operations, accounting for energy consumption,

fuel, water use and waste. This allows the researcher to obtain conclusive graphs that summarize the impact in order to intervene in a targeted way. The main source of emission for the planned intervention is due to the replacement of windows and doors, followed by floors and horizontal slabs and the facade cladding. The construction phase (A5) has a limited weight (16.7%), while the gate-to-site transport phase produces less than 2% of emissions: an excellent result obtained with a reduction in travel (larger and full-capacity trucking load) and providing supplies as local as possible (Figure 10).

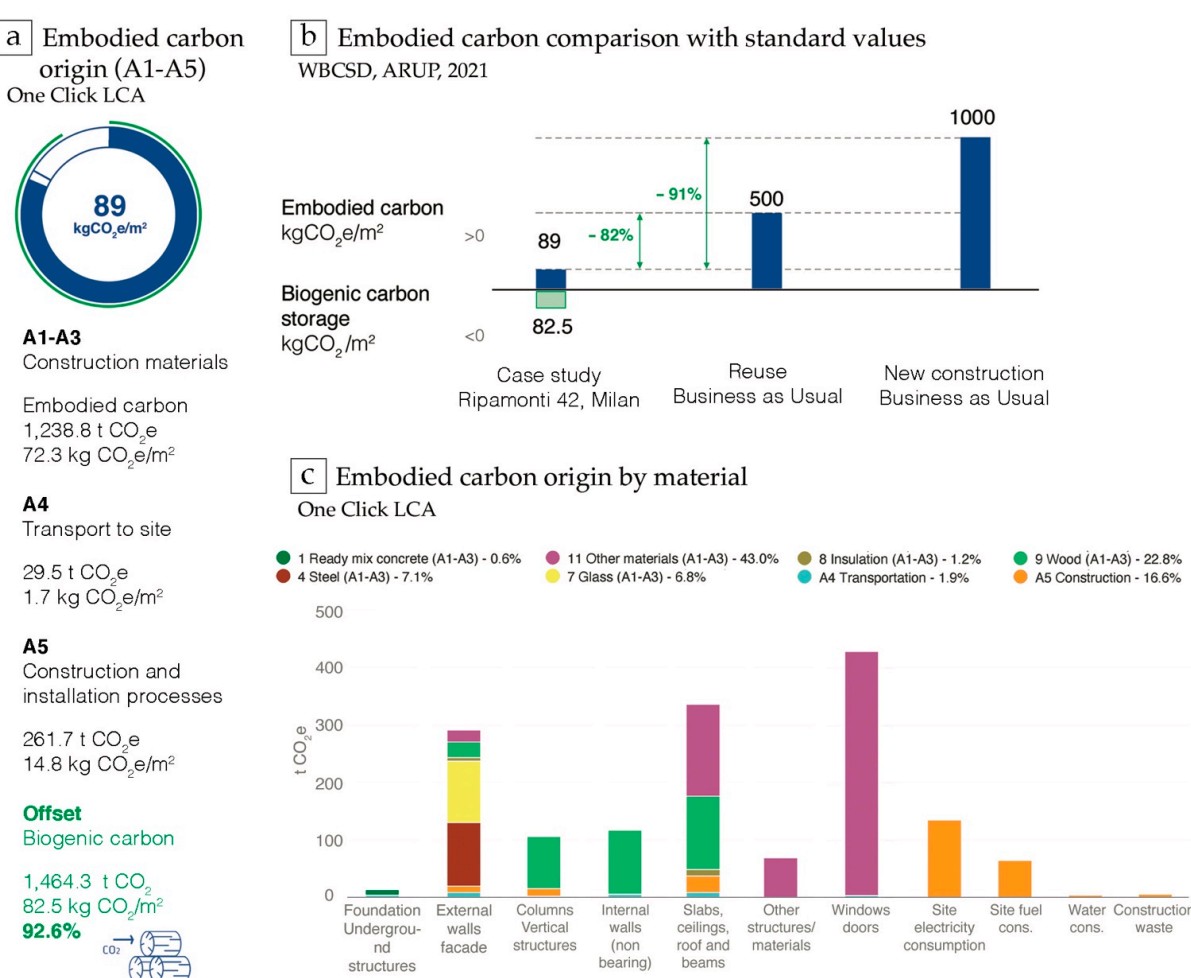

**Figure 10.** (**a**) Values of embodied carbon obtained with One Click LCA software for the case study; (**b**) the case study performs well compared to the standard embodied carbon values [29]; (**c**) embodied carbon emissions divided by material; bar chart obtained with One Click LCA software.

Table A1 in Appendix A shows the values entered in the software and on the Microsoft Excel spreadsheet among which the quantities of the components (increased by 10% to have a safety margin in order to balance possible exceeding emissions during the building process) and the carbon stored by the biogenic materials (referred to the effective quantity of material, without the 10% increase), with which the total quantity stored in the building was obtained.

Part A reports, for each technological system, the building material or component, the quantity and the relative unit of measurement, the reference to the position in the building, the distance from the production site, the type of road transport (proposed from the software in most cases), which tries to effectively evaluate how the delivery of materials could take place, the waste expected to put the product into operation and the quantities of embodied carbon and biogenic carbon stored. Part B focuses on site operations; in this case, most of the operations will be carried out dry for the retrofit of the existing building, a construction site with work vehicles such as cranes and excavators must be provided for the landscape

arrangements of the outdoor areas and for the construction of the new tower. The calculation considered the energy consumption of electricity, estimated at 15 kWh/m$^2$ for a dry construction site starting from the average data provided by the software of 25 kWh/m$^2$, the amount of water consumed and the amount of diesel fuel to power the construction equipment.

A relevant point of the calculation concerned construction and demolition waste (C&DW) transport and management. The retrofit work will produce large quantities of waste, that can, however, mainly be recycled; in fact, according to EU Waste Framework Directive (2008/98/EC), by 2020 a target of 70% of weight is set for the recycling of non-hazardous C&DW materials [57]. In this particular case, the facade systems (glazed panels and spandrel) can be addressed to the almost total recovery of aluminum and glass, the internal dry partitions and the Masonite doors, made of Masonite and aluminum, with plasterboard panels, can be recycled (aluminum) or undergo a downcycling process (Masonite), the internal masonry partitions of the plate volume as well as the precast concrete infill panels will have to be disposed of in landfills or recovered as aggregates for new concrete castings (foundations) or buried as filling in the external parts of the lot, for example in correspondence with the ramps and stairs connecting via Ripamonti. The materials to be recycled or disposed of in landfills must therefore be transported outside the lot, generating additional emissions that are included in the calculation.

### 3.3. Net Zero Carbon Buildings: Adopted Strategies

The results shown in Sections 3.1 and 3.2 obtained for the Ripamonti 42 asset were possible thanks to the adoption of several strategies to reduce embodied carbon emissions.

To reduce embodied carbon, the project follows the line of reusing existing buildings, increasing the useful surface by about 25% ("Build Less, Build Clever") [15]. The added volumes are very flexible and adaptable over time, thanks to the generous story height and very regular structures located on the perimeter to allow free internal distribution [36]. The materials are conceived to be almost entirely produced in Italy and come from a radius of about one hundred kilometers, with all of them coming from within five hundred kilometers. The principles of the circular economy were considered in the choice of materials, favoring recycled and recyclable products without causing downcycling of raw materials. For this reason, the construction details have been designed with the purpose of disassembling the components, mostly with dry connections to follow the principles of design for adaptability and design for disassembly [32]. To compensate for the remaining embodied carbon biogenic materials were chosen, in particular, wood for the wall and frame structures, cork for the insulating panels and some internal cladding surfaces, wooden and aluminum frames for the new windows; this contribution makes it possible to approach almost entirely the goal of carbon neutrality. To reduce operational carbon emissions, it is planned to adopt solutions for passive design, such as natural ventilation and lighting, passive heating and cooling and adaptive comfort theories: the opening of large suitably screened windows allows passive heating indoor spaces in winter and cooling in summer by natural ventilation [24].

The building must in any case be equipped with systems for mechanical ventilation and air treatment, as well as the production of domestic hot water. To compensate for these needs, the adoption of a system with geothermal heat pumps, located in the second basement, connected to air distribution systems, is planned. Open circuit geothermal probes can be installed underground in correspondence with the internal spaces, on which it is already planned to excavate for the purpose of greening and the new landscape design. This solution is also adopted on a neighborhood scale for the Scalo Romana masterplan, as it is very efficient in areas where the aquifer is superficial [52].

The electricity necessary for the functioning of the complex can be produced mainly on site thanks to the installation of a photovoltaic park on the green roofs of the building, together with the integration of mimetic photovoltaic panels in the ventilated façade that covers the south elevation of the plate and the hinged volumes. Regarding emissions due to the movement of occupants, which do not strictly depend on the building, the reduction

of car travel for employees and visitors is encouraged by the use of both public transport and soft mobility, both by reducing the stalls for private parking inside the lot. In fact, compared to the starting configuration with floors 0, −1 and −2 dedicated for parking, the project envisages leaving only part of the second basement floor for this function and installing charging systems for electric mobility, while on the first basement floor a bicycle parking has been created, connected directly with offices and research center: a choice in support of the healthy cities theme and active mobility [41].

## 4. Discussion

This study investigates the application of the principles for limiting carbon emissions in the redevelopment process of an office building in Milan, in order to comply with the Net Zero Carbon Building requirement, or an overall zero emissions balance during the building's life cycle. To demonstrate this goal, it is necessary to develop a WLCA analysis of the intervention [29]. For the purposes of the study, it was of main interest to focus on the embodied carbon emissions, since the methods for reducing the operating contributions on $CO_2$e emissions in construction are more widely known [58]; in this case the principles of passive design [15–23] and the production of energy from renewable sources on site [39] have been applied.

The calculation of the embodied carbon was carried out thanks to the One Click LCA software, with which it was possible to estimate the emissions related to the intervention at 89 kg $CO_2$e/m$^2$ for the A1-A5 modules (production of materials, transport and construction), which is a very low value when compared with the averages for new constructions and retrofits, that are 1000 kg $CO_2$e/m$^2$ for new constructions and 500 kg $CO_2$e/m$^2$ in the cases of reuse and redevelopment [29]. This result is possible thanks to the opportunity of maintaining most of the existing building structure; moreover, the new additions are made with wood structures and bio-based materials for insulation and finishes, guaranteeing a reduced carbon impact compared to the traditional materials (concrete, steel, screed etc.) and a significative carbon storage during the production phase of the natural materials. The negative contribution given by natural materials has been counted separately, resulting in an almost neutral balance of emissions. However, these emissions could be re-emitted during the disposal of materials in the EoL phase if they were not to be carefully treated [37,44–46].

Therefore, an additional investigation would be needed on how to calculate embodied carbon also associated with modules B1-B5 and C accurately and reliably, although it is only a fraction of the emissions in module A, in order to have an overall picture of the entire life cycle of the building. It is important to note that great attention and further research must be addressed to the realization phase in compliance with the specific materials supply and construction site management requirements expressed in this paper, since the choice of materials (both for production methods and origin) can have a significant impact on the veracity of the as-built calculations.

## 5. Conclusions

The paper focuses on the theme of Net Zero Carbon Buildings (NZCB) as a possible solution to decarbonize the building sector, which is actually the main contributor of greenhouse gases emissions (38%) which have been scientifically proven to increase temperature and to be the main cause of climate change [2,13,15].

Firstly, the design process of a NZCB must include an evaluation of embodied carbon emissions due to production, supply, transport of materials, construction, maintenance and end of life since the early stages, in order to minimize each carbon contribution. In the meanwhile, a series of strategies should be evaluated, aiming at cutting emissions thanks to a reduction or a cleverer output in consumption of new materials, with a preference for reuse and recycled when possible. Embodied carbon can be assessed using specific LCA software according to the EN 15978:2011 standard, sourcing the data from the products' EPD and the bill of quantities (BOQ) derived from the BIM model of the designed building.

Secondly, the paper focuses on the application of these principles to a case study of an abandoned building in Milan, Italy, paving the way to future applications, since NZCB has not yet been introduced in the country legislation; this is still a barrier when sourcing data from EPDs and for the lack of built examples. Performing the analysis with the software One Click LCA, a result of 91% reduction in embodied carbon emissions was obtained, while the choice of bio-based materials for new structure and insulation guaranteed sufficient biogenic carbon storage to compensate for embodied carbon emissions.

The lesson learnt developing the project in the Milan context can be helpful for future studies and applications. It was possible to satisfy the NZCB goal only with the complete reuse of the structure, with narrow demolitions and with volumetric additions made of bio-based materials, while the flexibility and adaptability in the future extended the life of the building and limited the necessity for big building interventions to adapt to new needs.

The growing relevance of the NZCB standard in the future can probably become a quality indicator of an asset, proving its environmental sustainability, and thus gaining a better market visibility and keeping better value over time. Nevertheless, this carbon-centric approach is quite different from the common practice, requiring specific training for both designers and construction managers; as well, the low interest in building decarbonization by legislative requirements constitutes a second main barriers to the NZCB implementation.

Once a proper legislative framework defines for each country the performance indicators to assess that the as built is a NZCB, contractors and developers will be more confident in facing the transition towards a new way of developing architecture, for the benefit of present and future generations, to comply with the 2050 goal of complete decarbonization in the building sector set by UNFCCC [11].

Meanwhile further research on this topic is needed, especially to pre-announce more solidly the outcome of modules B, C and D of the life cycle assessment, increasing the quality of information about the building itself.

**Author Contributions:** Conceptualization, D.B. and D.T.; methodology, D.B. and D.T.; software, D.T.; resources, D.B. and D.T.; data curation, D.T.; writing—original draft preparation, D.B. and D.T.; writing—review and editing, D.B. and D.T. All authors have read and agreed to the published version of the manuscript.

**Funding:** This research received no external funding.

**Institutional Review Board Statement:** Not applicable.

**Informed Consent Statement:** Not applicable.

**Data Availability Statement:** The data presented in this study are available on request from the corresponding author.

**Conflicts of Interest:** The authors declare no conflict of interest.

## Abbreviation

| | |
|---|---|
| BIM | Building Information Modeling |
| BOQ | Bill of Quantities |
| C&DW | Construction and Demolition Waste |
| DT | Digital Twin |
| EoL | End of Life |
| EPD | Environmental Product Declaration |
| GFA | Gross Floor Area |
| LCA | Life Cycle Assessment |
| NZCB | Net Zero Carbon Buildings |
| nZEB | nearly Zero Energy Buildings |
| WLCA | Whole Life Carbon Assessment |

# Appendix A

**Table A1.** Embodied carbon assessment for the Ripamonti 42 building. Each material has its own value expressed both in terms of GWP and Biogenic carbon.

**Embodied Carbon Report—Construction Materials and Construction Site Operations—One Click LCA Planetary Italy**

| A—Construction materials | Quantity | Unit | | | km | Transport Mode | Waste (%) | GWP [kgCO$_2$e/unit] | Biogenic CO$_2$ [kgCO$_2$/unit] |
|---|---|---|---|---|---|---|---|---|---|
| **1. Foundations** | | | | | | **16 t CO$_2$e—1%** | | | |
| **Foundations, subsoil, basements and retaining walls** | | | | | | | | | |
| Ready-mix concrete, normal strength C28/35 60% GGBS content | 86.85 | m$^3$ | | | 100 | Concrete mixer truck | 4 | 153.89 | 1.01 |
| Reinforcement steel (rebar), 80% recycled content | 0.60 | t | | | 50 | Trailer combination, 40 ton, 100% fill | 5 | 920.00 | 0.00 |
| Formwork panel from EPS for crawl s space (Aircrab H35) | 143 | m$^2$ | | | 50 | Large delivery truck, 9 ton, 50% fill | 4 | 8.00 | 0.00 |
| PVC waterproofing membrane, 1.5 mm (Mapelan plus) | 154 | m$^2$ | 1.5 | mm | 50 | Large delivery truck, 9 ton, 100% fill | 10 | 4.76 | 0.00 |
| **2. Vertical structures and facades** | | | | | | **515 t CO$_2$e—33%** | | | |
| **External walls and facades** | | | | | | | | | |
| Natural cork insulation panel, λ = 0.043 W/mK (LIS srl) | 5.950 | m$^2$ | 100 | mm | 350 | Large delivery truck, 9 ton, 100% fill | 8 | 0.87 | 30.43 |
| Float glass, single pane, generic | 5.500 | m$^2$ | 6 | mm | 100 | Large delivery truck, 9 ton, 100% fill | 1 | 12.25 | 0.00 |
| Aluminium frame windows, U = 1.0 W/m$^2$K (Metra) | 93 | m$^2$ | | | 100 | Trailer combination, 40 ton, 50% fill | None | 159.00 | 0.00 |
| Solid wood panels, 493 kg/m$^3$, 9% moisture content (Nordpan) | 108 | m$^3$ | | | 500 | Large delivery truck, 9 ton, 100% fill | 18 | 256.83 | 903.83 |
| Sandstone cladding, natural, 20 mm, (Casone group) | 43 | m$^3$ | | | 60 | Large delivery truck, 9 ton, 50% fill | 5 | 151.47 | 0.00 |
| Steel sheets, generic, 90% recycled content S235 | 12.50 | m$^3$ | | | 50 | Trailer combination, 40 ton, 100% fill | 3 | 4783.32 | 0.00 |
| **Pillars, columns, structural walls** | | | | | | | | | |
| Glued laminated timber (Glulam), 464 kg/m$^3$, 12% moisture (Rubner) | 37.60 | m$^3$ | | | 400 | Trailer combination, 40 ton, 100% fill | 17 | 204.67 | 850.67 |
| Cross laminated timber (CLT), 461 kg/m$^3$, 11% moisture (Rubner) | 390 | m$^3$ | | | 100 | Trailer combination, 40 ton, 100% fill | 17 | 181.17 | 845.17 |
| Timber frame external wall element, U = 0.16 W/m$^2$K (Lapwall) | 1.032 | m$^2$ | | | 500 | Trailer combination, 40 ton, 100% fill | None | 13.00 | 23.90 |
| **Internal walls and non-structural elements** | | | | | | | | | |
| Timber frame external wall element, U = 0.16 W/m$^2$K (Lapwall) | 5.287 | m$^2$ | | | 400 | Trailer combination, 40 ton, 100% fill | None | 21.20 | 17.30 |
| **3. Horizontal structures: beams, slabs, roofs** | | | | | | **337 t CO$_2$e—21%** | | | |
| **Slabs, ceilings, beams and roofs** | | | | | | | | | |
| Cross laminated timber (CLT), 461 kg/m$^3$, 11% moisture (Rubner) | 626 | m$^3$ | | | 100 | Trailer combination, 40 ton, 100% fill | 17 | 181.17 | 845.17 |
| Glued laminated timber (Glulam), 464 kg/m$^3$, 12% moisture (Rubner) | 69.50 | m$^3$ | | | 400 | Trailer combination, 40 ton, 100% fill | 17 | 204.67 | 850.67 |
| NHL (natural hydraulic lime) based floor screed (Domus VR opus-c) | 2.600 | m$^2$ | 80 | mm | 100 | Large delivery truck, 9 ton, 100% fill | 13 | 149.53 | 0.00 |
| Natural cork insulation panel, lambda = 0.043 W/mK (LIS srl) | 5.847 | m$^2$ | 100 | mm | 350 | Trailer combination, 40 ton, 100% fill | 8 | 0.87 | 30.43 |
| Wet sand (Gruppo Bassanetti) | 321 | ton | | | 40 | Dumper truck, 19 ton, 100% fill | None | 0.00 | 0.00 |
| Wood-alu frame window, triple-glazed, U = 1.0 W/m$^2$K (Pihla group) | 500 | m$^2$ | | | 100 | Delivery van, 1.2 ton, 100% fill | None | 118.81 | 24.33 |
| Aluminium frame windows, U = 1.0 W/m$^2$K (Metra) | 350 | m$^2$ | | | 100 | Trailer combination, 40 ton, 100% fill | None | 159.00 | 0.00 |
| Bitumen-polymer membrane roofing, 2 layer (EWA) | 3.450 | m$^2$ | 5 | mm | 90 | Trailer combination, 40 ton, 100% fill | 10 | 5.45 | 0.00 |
| EPS insulation, L = 0.036 W/mK, 36 mm (Rexpol) | 3.450 | m$^2$ | | | 90 | Trailer combination, 40 ton, 100% fill | 4 | 1.95 | 0.00 |

**Table A1.** *Cont.*

| Embodied Carbon Report—Construction Materials and Construction Site Operations—One Click LCA Planetary Italy | | | | | | | | |
|---|---|---|---|---|---|---|---|---|
| **A—Construction materials** | **Quantity** | **Unit** | | **km** | **Transport Mode** | **Waste (%)** | **GWP [kgCO$_2$e/unit]** | **Biogenic CO$_2$ [kgCO$_2$/unit]** |
| **4. Other structures and materials** | | | | | **499 t CO$_2$e—32%** | | | |
| **Other structures and materials** | | | | | | | | |
| Elevator, 630 kg capacity, for passenger use (Monospace 500 DX-Kone) | 8 | unit | | 320 | Large delivery truck, 9 ton, 100% fill | None | 8529.84 | 23.17 |
| **Windows and doors** | | | | | | | | |
| Wood-alu frame window, triple-glazed, U = 1.0 W/m$^2$K (Pihla group) | 3.180 | m$^2$ | | 100 | Delivery van, 1.2 ton, 100% fill | None | 118.81 | 24.33 |
| Aluminium frame windows, U = 1.0 W/m$^2$K (Metra) | 80 | m$^2$ | | 100 | Large delivery truck, 9 ton, 100% fill | None | 159.00 | 0.00 |
| MDF hollow-core door, veneered, (Abet Laminam) | 346 | unit | | 100 | Trailer combination, 40 ton, 100% fill | None | 102.70 | 10.09 |
| **B—Construction site operations** | **Quantity** | **Unit** | **CO$_2$e** | **km** | **Transport Mode** | **Waste (%)** | **GWP [kgCO$_2$/unità]** | |
| **1. Construction site scenarios** | | | | | | | | |
| **2. Energy use on the site** | | | | | **200 t CO$_2$e—13%** | | | |
| **Site electricity consumption** | | | | | | | | |
| Electricity, Italy | 267.000 | kWh | 135 t—9% | 15 kWh/m$^2$ | | | 0.51 | |
| **Site district heating consumption** | | | | | | | | |
| **Site fuel consumption** | | | | | | | | |
| Diesel | 20.000 | L | 65 t—4% | | | | 3.24 | |
| **Machine hours** | | | | | | | | |
| **3. Materials use on the site (that do not constitute part of the asset)** | | | | | | | | |
| **Material use** | | | | | | | | |
| **4. Water use on the site** | | | | | **3 t CO$_2$e** | | | |
| **Water consumption** | | | | | | | | |
| Tap water, clean and wastewater | 5.000 | m$^3$ | 3.5 t—0.2% | | | | 0.69 | |
| **5. Waste generated on the site** | | | | | **5 t CO$_2$e** | | | |
| **Construction waste** | | | | | | | | |
| Inert waste landfilling | 450 | ton | 4.8 t—0.3% | 0 | Trailer combination, 40 ton, 100% fill | None | 0.01 | |
| **6. Additional trips for the transport to the construction site** | | | | | **0.51 t CO$_2$e** | | | |
| **Additional transportation** | | | | | | | | |
| Transported mass | 346 | ton | 0.51 t—~0% | 20 | Dumper truck, 19 ton, 100% fill | None | | |

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
