# Peer review of "Reuse and Retrofitting Strategies for a Net Zero Carbon Building in Milan: An Analytic Evaluation"

_sustainability, doi:10.3390/su142316115_

Round 1
Reviewer 1 Report
The reviewed manuscript is at an average level in terms of the originality of the solutions presented and the final conclusions formulated, but it will probably meet with the interest of readers, and its results can be used in projects related to the reuse and modernization of existing buildings to the Net Zero-Carbon Building standard (NZCB).
The manuscript is primarily descriptive, but it is clear and documented. One Click LCA software was used to assess emissions according to the Whole Life-cycle Carbon Assessment (WLCA) methodology. The procedures for recovering the building stock (addition, subtraction, wrapping) presented in the case study (Milan building) will be particularly suitable for planning the reuse of existing buildings.
The manuscript requires minor changes, especially when it comes to linguistic and stylistic correctness. It is also recommended to introduce a nomenclature and list of abbreviations used, which will significantly improve the readability of the article.
Author Response
Dear reviewer,
thank you very much for the precious comments and suggestions.
Please find below the point-by-point response to your review.
- A critical interpretation of the solutions applied for the case study has been added in paragraphs 3.1 and 3.2
- The strategy solutions for the reuse of existing building (addition, substraction, wrapping) has been introduced in a more general way in paragraph 2.1
- A nomenclature and a list of abbreviation has been introduced at the end of the paper.
Best regards
Reviewer 2 Report
This paper uses a case study to investigate feasible strategies and design principles for minimizing carbon emissions during the lifecycle of redeveloped buildings. The topic of this paper is interesting and deserves investigation. However, there is a lack of discussion on research methodology, research details, and interpretation of the results.
Overall, it is not clear what the study is presenting or contributing to the body of knowledge. The novelty and originality of the presented work are very weak. The aim of this research is ambiguous – investigation of the application of design principles for limiting carbon emission, and this aim limits readers to clearly understand research results and potential implications. Furthermore, the methodologies used in the paper are less robust, and their narrative and details are not clearly presented. Moreover, the manuscript content does not sound scientific and shows limitations in delivering new and significant information compared to existing research on LCC analysis of buildings. More comprehensive studies are needed to verify, while emphasizing its novelty.
The paper does not show a critical understanding of the relevant literature. The literature review is not complete and is scattered in Chapters 2 and 3, which reduces readability. Authors also should separate literature review contents from the research method and results and include recent literature on LCC analysis of buildings as well as its BIM-based application.
Please explain more details about the methodologies used in this research. For instance, there are no technical details (e.g., main concept, equation, key parameters) about the chosen WLCA method out of the ones suggested by diverse researchers, governments, or institutions. How to use BIM for LCC analysis on One Click LCC need to be discussed (data modelling, information requirement, data mapping). In addition, an overview of the research process needs to be presented to provide a better understanding of this research design.
This work lacks to convey the research result clearly. The diverse design strategies/principles were applied to the projects, but their descriptions are narrative and non-technical. The impact on the reduction of carbon emissions depends on the specification of each strategy/principle and their interaction. The current result could be used to identify high-ranked strategies/principles, but its evidence is not sufficiently provided. To produce sound implications, the details of analyzed strategies/principles should be discussed, and the interdependency among strategies/principles in reducing carbon emissions need to be analyzed.
Abstract should summarize the main findings and provide implications or directions for future work.
This work still lacks the discussion part. It would be better to discuss more on the application and results obtained from this research. What are the potential implications for different communities (researchers, contractors, developers)? What is the scientific significance of this research outcome compared to other past related works?
The authors need to add Figure 2 and a colour legend for Figure 4 (a).
Author Response
Dear reviewer,
thank you very much for the precious comments and suggestions.
Please find below the point-by-point response to your review.
- A discussion on research methodology, research details, and interpretation of the results has been introduced in the paper. In particular the Material and methods have been revised to add more details and structured to introduce firstly the strategies of reuse, retrofit and for Net Zero Carbon Buildings, and secondly how these are applied into the case study.
- The research results and potential implications are presented in the results paragraphs 3.2 and 3.3 and in the discussion and conclusion paragraph.
- Technical details have been described in paragraph 2.6 and in the Results section.
- References have been implemented
- Technical details (main concept, equation, key parameters) chosen in WLCA method has been added in the Results section
- A discussion about the use BIM for LCA analysis on One Click LCA has been presented in paragraph 2.5
- The strategy solutions for the reuse of existing building have been described in a more technical way and not only as application on a specific case study in paragraph 2.1
- Abstract and Conclusions have been totally revised
- The Discussion of the paper has been implemented
- Figure 2 and a color legend for Figure 4 are now in the paper
Best regards
Reviewer 3 Report
The article appears more like a report about the specific design choices rather than a scientific paper. The design strategies adopted should be described distinguishing the local and specific constraints from a more general approach and method to be possibly applied in other cases. Maybe the explanation of the spatial as well as technological design choices as the result of a discussion among different available options could help deepening the methodological articulation of the work.
The goal of the authors is “aimed at developing a refurbishment project of an abandoned building by quantitatively evaluating the impact of the embodied carbon associated with the intervention, with the aim of being a Net Zero Carbon Building”. The research question is not clearly stated. A possible improvement could be to consider the case study discussed as a methodological means to offer a scientific contribution to the larger issue of addressing the problem o NZCB design applied to refurbishment interventions.
Although the topic is relevant and tackles with the issue of Net zero carbon building, a yet unexplored topic from the design methodology and approaches perspective, the paper mainly focuses on the specific design experience developed and draws limited and too implicit results to be applied on other projects or to contribute to the theoretical research.
However, considering research through design a significant means of scientific advancement, a more structured description of the lesson learnt through the specific design experience could considerably improve the contribution.
The methodological development of the paper could be enhanced by clearly stating which are the choices stemming from the specific constraints and goals of the case study and which are the approaches and methods to be scaled-up and applied in other case studies.
Moreover, the conclusions could be considerably implemented accordingly, as the related dissertation is currently too generic.
The references could consequently be implemented deepening the specific subject of the NZCB design.
Finally, minor details concerning the design brief are taken for granted and Figure 2 does not appear in the pdf.
Author Response
Dear reviewer,
thank you very much for the precious comments and suggestions.
Please find below the point-by-point response to your review.
- The strategy solutions for the reuse of existing building have been described in a more technical way and not only as application on a specific case study in paragraph 2.1
- A more precise description about technological design choices has been introduced in paragraphs 3.1 and 3.2, while the general strategies are described in paragraphs 2.1 to 2.5
- The NZCB design applied to refurbishment interventions has been presented in a clearer way (paragraph 3.1)
- The conclusion and the discussion of the paper have been implemented with theoretical content about NZCB design in order to set the case study in the wider scenario of building reuse and retrofit strategies, so that they can be applied on other projects.
- The lesson learnt through the specific design experience has been added to the paper, in particular in the Conclusions section
- References have been implemented
- Figure 2 is now presented in the paper
Best regards
Round 2
Reviewer 2 Report
The revision has addressed significant issues and is much improved. Thanks for the authors' efforts. There are only a couple of issues:
1) Abstract and Introduction. The aim of this research is ambiguous due to the absence of its clear statement. One sentence which can clearly state the research aim should be placed in the abstract and introduction.
2) Research Method. As the authors addressed, there are multiple tools for analyzing LCA and WLCA. The reason why OneClick LCA is selected in this study should be justified since the LCA method embedded in this tool determine the accuracy of the research result.
The concept of BIM (model or modelling process) in this paper should be discussed. The level of development (LOD) of the created BIM data of the case study building.
3) Use of abbreviation.
· EPD: used before the statement of its full phrase in line 324 (in abstract, line 320)
· NZCB, NZCB: full phrase are used after abbreviation definition in line 143
Author Response
Dear reviewer,
thank you for the careful and precise feedback on the paper.
We have worked to improve the contribution focusing on the points you highlighted. In particular in the Abstract and in the Introduction we have cleared the aim of the research and we have added more information about the choice of using the BIM model and the specific LCA tool.
To conclude, we have checked the use of abbreviation and we have highlighted in yellow the main modifications (in the track changes version).
Best regards
Reviewer 3 Report
The paper has undergone a significant implementation and has taken in consideration the suggestions indicated, also as hints for the integration of further information and a clearer structure of the methodology. The conclusion could be perhaps improved through a better statement of the limits and barriers of the change in approach towards the NZCB goal. A final editing to improve the English language might be necessary.
Author Response
Dear reviewer,
thank you for the careful and precise feedback on the paper.
We have worked to improve the contribution focusing on the points you highlighted. In particular we have widened the Conclusion discussing about limits and barriers of NZCB transition in the building construction sector.
To conclude, we have checked the English language throughout the paper and we have highlighted in yellow the main modifications (in the track changes version).
Best regards